# Early Warning of AC Salt Fog Flashover on Composite Insulators Using Fiber Bragg Grating Sensing and Visible Arc Images

**DOI:** 10.3390/mi16101171

**Published:** 2025-10-16

**Authors:** Xiaoxiang Wu, Yanpeng Hao, Zijian Wu, Jikai Bi, Haixin Wu, Lei Huang

**Affiliations:** 1School of Electric Power Engineering, South China University of Technology, Guangzhou 510640, China; 2School of Electrical and Information Engineering, Zhengzhou University, Zhengzhou 450001, China; 3China Southern Power Grid Research Technology Co., Ltd., Guangzhou 510663, China

**Keywords:** fiber Bragg grating, fiber-optic composite insulator, interface temperature, visible discharge images, flashover early warning

## Abstract

External insulation of coastal power grids faces harsh conditions and is highly susceptible to flashover. Currently, technologies for online monitoring and flashover early warning are severely lacking. As a reflective passive sensing device, Fiber Bragg Grating (FBG) enables the monitoring of surface discharge and provides an early warning for flashover on external insulation. A 10 kV fiber-optic composite insulator was developed in this study. A linear relationship between the FBG central wavelength and interfacial temperature was established through temperature calibration experiments. Coastal salt fog conditions were simulated in an artificial fog chamber, where AC pollution flashover tests were performed on the fiber-optic composite insulator. FBG central wavelength and visible images of discharge were synchronously acquired during experimentation. Experimental results indicate that the interfacial locations on FBGs where the temperature increases during flashover coincide with the positions of visible discharge arcs, demonstrating the effectiveness of the monitoring method. A temperature rise rate of 4.88 × 10^−2^ °C/s was found to trigger flashover warning, while a rate of 4.96 × 10^−2^ °C/s initiated trip protection. A discharge-region ratio characteristic was proposed for visible discharge images based on highlight area ratio, R-channel deviation, and mean saturation features. This characteristic serves as a flashover warning when its value reaches 46.7%. This study provides a novel research approach for online monitoring and flashover early warning of external insulation in coastal salt fog environments.

## 1. Introduction

In recent years, offshore wind power has emerged as a critical component in energy transition due to its inherent advantages [1]. Composite insulators, valued for their unique properties such as exceptional hydrophobicity and pollution flashover resistance of silicone rubber materials, have been widely deployed in coastal transmission lines [2,3]. However, high humidity levels and elevated salt content in sea wind or fog increase the risk of insulator flashover [4]. Salt fog flashovers can lead to large-scale prolonged power outages, causing severe impacts on electrical grids [5]. Optical fiber sensing and image-processing technologies enable online monitoring of operating conditions for composite insulators. Understanding discharge development mechanisms through these methods is of significant value in preventing insulation flashover [4,6,7,8,9].

Electrical equipment faces long-term operating times in high-voltage and strong electromagnetic environments, imposing stringent requirements on electromagnetic compatibility and insulation performance of photoelectric sensors. FBG offers advantages including compact size, excellent insulation, corrosion resistance, immunity to electromagnetic interference, and passive sensing. These capabilities have led to their extensive research and application in fields such as aviation, energy, traffic, and power systems [10,11,12,13,14,15].

Fiber-optic composite insulators incorporate FBGs within their structure, enabling real-time monitoring of operational status. In 2004, Trouillet A. et al. [16] bonded an FBG to the core rod surface of a composite insulator using epoxy resin, matching the rod’s base material. Calibration experiments revealed a linear relationship between FBG central wavelength shift and both strain and temperature, with a strain sensitivity of 1.18 pm/με and temperature sensitivity of 17 pm/°C. In 2009, Kerrouche, A. et al. [17] proposed embedding FBGs into axial grooves on the surface of the insulator core rod to monitor axial strain. Experimental results demonstrated strong agreement among strain measurements from this method, calibrated tensile machine data, and results from resistive strain gauges on the core rod. In 2011, Cai W. et al. [18] monitored temperature and stress states within fiber-optic composite insulators and conducted grid-connected electrical and mechanical tests. The results demonstrated that such insulators can effectively enable the real-time monitoring of core rod temperature and stress. In 2016, Chen W. et al. [19] proposed a mechanical numerical model for fiber-optic composite insulators. This model revealed relationships among crimping stress in fittings, ambient temperature, and spectral response of FBGs, with its validity confirmed under experimental conditions. In 2022, Hao Y. et al. [20] removed a 36 mm × 7 mm sheath segment from a composite insulator. A molybdenum wire simulating an interfacial heat source and a FBG were bonded to the exposed core rod surface. The sheath segment was then repositioned, and an additional FBG was attached to the outer sheath surface to monitor surface temperature. By applying a DC current to the molybdenum wire to generate heat, the relationship between abnormal interfacial temperature rise and both interfacial and surface temperatures was investigated. In 2022, they embedded a 3 × 4 array of quasi-distributed FBGs into the core rod interface of a composite insulator to monitor surface dry-band arcing under wet and contaminated conditions. It was found that FBGs enable limited online monitoring of both instantaneous and short-duration arcs [21]. In 2023, by implanting FBGs into the interface between silicone rubber and an epoxy resin plate, a method was proposed for monitoring dynamic deformation of water droplets and flashover discharge on hydrophobic insulation surfaces based on interfacial temperature measurements [22].

The aforementioned research has focused primarily on stress and temperature monitoring in fiber-optic composite insulators. Visible-light images of insulator flashover discharge contain substantial feature information, such as arc morphology, color, and length. In 2016, Fang C. et al. [23] conducted AC pollution flashover tests on porcelain insulators and extracted characteristics related to arc area. In 2021, Li F. et al. extracted and analyzed R, G, and B component values within arc regions of needle-plane corona discharge images. A characteristic parameter based on the difference in ratios between B and G components was proposed, which can effectively distinguish different stages of discharge development [24]. Also in 2021, Thanh P.N. et al. developed a multi-parameter model incorporating ambient temperature, humidity, dew point, spark discharge area ratio, and luminance variation. A significant correlation was identified between spark discharge and leakage current [25].

Distinct from prior studies that primarily utilize optical fiber sensing or visible-light imaging in isolation, this work introduces a substantively novel methodology through the deep integration of both technologies. Moving beyond simple parallel data collection, we establish a cooperative decision-making logic where the FBG-measured interfacial temperature rise serves as the primary trigger, which is then critically verified by visible-light arc characteristics. This synergistic approach effectively overcomes the inherent limitations of single-parameter methods.

This study manufactured a 10 kV fiber-optic composite insulator and discusses the temperature measurement principles of FBG sensing. AC flashover tests were conducted in an artificial salt fog chamber, with interfacial temperature characteristics analyzed and features from arc visible-light images extracted during flashover. Experimental results demonstrate that the fiber-optic composite insulator enables effective online monitoring and early flashover warning.

## 2. Temperature Detection Principle of FBG

A typical optical fiber features a coaxial cylindrical structure, consisting of a core, cladding, and coating layer sequentially from interior to exterior [26], as shown in Figure 1.

The FBG is an optical fiber segment exhibiting periodic modulation of the core refractive index. This periodic refractive index distribution enables the wavelength-selective reflection and transmission of light in the grating. As a reflective grating, its initial central wavelength is given by Equation (1) [27]:(1)$$\lambda =2n_{eff}\cdot \Lambda_{T}$$ In this equation, $n_{eff}$ represents the effective refractive index of the fiber grating, which is a constant typically within the range of 1.446 to 1.485, with the specific value provided by the FBG manufacturer’s datasheet, and Λ denotes the grating period.

Temperature variation induces wavelength drift in FBGs due to thermal expansion, thermo-optic, and elasto-optic effects. The complete relationship between temperature and wavelength is given by Equation (2) [28]:(2)$$\frac{\Delta \lambda }{\lambda }=\left\{\frac{1}{n_{eff}}\left[\xi -\frac{n_{\mathrm{eff}}^{3}}{2}\left(P_{11}+P_{12}\right)\alpha +k_{wg}\alpha \right]+\alpha \right\}\Delta T$$ In this equation, *ξ* represents the thermo-optic coefficient, *P*_11_ and *P*_12_ denote elasto-optic coefficients, α indicates the thermal expansion coefficient, and k_wg_ corresponds to the wavelength drift coefficient induced by waveguide effects. The wavelength sensitivity coefficient of a FBG is typically constant, resulting in highly linear wavelength output during temperature measurement.

The optical fiber material is fused silica, with *ξ* being 0.68 × 10^−5^/°C and α being 5.5 × 10^−7^/°C. Elasto-optic and waveguide effects exert minimal influence on FBG drift; therefore, Equation (2) can be simplified as follows:(3)$$\begin{array}{l} \frac{\Delta \lambda }{\lambda }=K_{T}\Delta T\\ K_{T}=\xi /n_{eff}+\alpha \end{array}$$

In summary, the central wavelength *λ* typically ranges from 1510 nm to 1590 nm, with variation not exceeding 1 nm, while the temperature coefficient *K_T_* fluctuates within 0.1%. These results demonstrate that embedding FBGs into composite insulators enables effective monitoring of interfacial temperature.

In this study, the thermal effect generated by the flashover arc is the dominant factor causing the sharp temperature rise at the interface, while potential accompanying mechanical strain influences are relatively transient and minimal. The early warning mechanism is based on the core criterion of the trend in the temperature rise rate, whose signal characteristics primarily reflect the continuous heat accumulation process. The above principle analysis and experimental results demonstrate a stable correlation between the FBG wavelength shift and temperature variation, where transient strain disturbances do not alter the overall evolution trend of the temperature rise rate. Consequently, focusing on the temperature parameter to establish the warning model in this research is reasonable and reliable.

## 3. Fabrication Method of Fiber-Optic Composite Insulators

Fabrication of fiber-optic composite insulators involves the following steps:(1)The epoxy resin–glass fiber core rod should be polished at first, to maximize adhesion strength, with positions for composite insulator sheds are marked on its surface.(2)Five optical fibers with Bragg gratings (customized by Zhongke Sensing Technology Co., Ltd., Shenzhen, China), the sensing length of each FBG is 10 mm.) were bonded to the core rod surface using epoxy (Kafuter K-703, Evergrande New Materials Technology Co., Ltd., Huizhou, China) adhesive with low shrinkage rate, high toughness, and excellent resistance to hygrothermal aging, aiming at effectively mitigating the internal stresses arising from the coefficient of thermal expansion mismatch among the Fiber-Reinforced Plastic (FRP), epoxy, and the optical fiber. The customized optical fiber with a total length of approximately 3.5 m was used. Among this, about 0.29 m was precisely positioned within the core rod-silicone rubber sheath interface of the composite insulator, serving as the strain-sensing region, while the remaining approximately 3.21 m served as the pigtail to ensure the measurement equipment could be located in a safe zone during high-voltage experiments. Fiber 1# and Fiber 4# were placed at identical positions to verify data accuracy. Layout of fiber gratings is shown in Figure 2a.(3)Room-temperature vulcanized (RTV) silicone rubber (Kafuter K-704, Evergrande New Materials Technology Co., Ltd., Huizhou, China) was uniformly applied to the core rod surface. The resulting core rod–fiber grating–RTV silicone rubber bonding structure is shown in Figure 2b.(4)Crimping of metal fittings was performed after initial RTV silicone rubber curing.(5)Following complete RTV silicone rubber curing and consistent with the standard manufacturing process for conventional composite insulators using High-Temperature Vulcanizing (HTV) silicone rubber encapsulation, the core rod was baked at a temperature of 140 °C for a duration of over 30 min. Final sealing was then carried out with adhesive to complete fabrication of the fiber-optic composite insulator, as shown in Figure 3a.

Structural parameters of the fiber-optic composite insulator are listed in Table 1. The detailed schematic diagram of each parameter is shown in Figure 3b.

## 4. Experimental Methodology

### 4.1. Testing System

A salt fog flashover testing system for fiber-optic composite insulators was constructed, as illustrated in Figure 4. In accordance with the standard GB/T 22707-2008 [29], test voltage was supplied by a single-phase transformer rated at 50 kV/250 kVA [30].

The fog chamber was constructed with six transparent acrylic panels, measuring 1.5 m × 1.5 m × 2 m. Power cables on the left side pass through wall bushings to connect to the plate specimen, while the right side incorporates an observation window and an inlet for the salt fog pipeline. A fogging device was placed outside the chamber, and salt fog was introduced through an inlet pipe equipped with uniformly spaced small holes for fog dispersion. A monitoring camera was mounted 1 m from the specimen. A FBG interrogator (model JEME-iFBG-S08, Tianlihua High Tech Co., Ltd., Wuhan, China) was employed to monitor interfacial temperature of the insulator, featuring a wavelength range of 1510–1590 nm, demodulation accuracy of 1 pm, wavelength sensitivity of 0.1 pm, and a maximum scanning frequency of 10 Hz. The ultrasonic atomization device model is JY-9.0F, and visible-light image obtained by Lenovo X5Q camera (Lenovo Group Ltd., Shenzhen, China).

The camera with a frame rate of 30 FPS was used in testing system due to the following reasons: (1) appropriate engineering cost; (2) it can better meet the needs of long-term monitoring compared to high-speed cameras; (3) the sampling interval of 30 FPS is 33.3 ms, but the duration of a single discharge event is about 60 ms [31], the camera’s principle is that any discharge activity occurring during the 33.3 ms exposure time will be recorded. Therefore, a camera with this frame rate can effectively track every discharge event.

According to IEC 60507 [32], prior to testing, sodium chloride and kaolin were weighed according to a salt deposit density of 0.1 mg/cm^2^ and an ash density of 1.0 mg/cm^2^. The contaminants were mixed with water to form a paste, uniformly applied to the specimen surface, and then vertically suspended in the fog chamber. A salt fog solution with conductivity of 3 mS/cm was used during tests. Before initiating the experiment, a timer was started to synchronously record visible-light images of arc discharge at various stages on the specimen surface along with central wavelengths of the fiber gratings.

### 4.2. Fiber-Grating Temperature Calibration Experiment

Temperature calibration of the FBG sensor on the specimen was conducted in a constant-temperature chamber. Chamber temperature was sequentially set to 0, 5, 10, 15, 30, 35, and 40 °C. After maintaining the plate at each target temperature for 8 h, the central wavelength of the grating was recorded, with a one-minute average value taken as the central wavelength at that temperature [33].

### 4.3. Experimental Procedure

Voltage across was increased steadily at test initiation. Upon first appearance and rapid extinction of an arc, voltage was held constant for data recording before resuming increase. This procedure was repeated until surface arcs developed into flashover. The voltage application process is illustrated in Figure 5.

A typical voltage escalation curve for AC salt fog flashover tests on fiber-optic composite insulators is shown in Figure 6. With increasing voltage, corona or partial discharge emerged on the insulation surface. Flashover trip occurred in the composite insulator when applied voltage reached the flashover level.

## 5. Results and Analysis

### 5.1. Fiber-Grating Temperature Calibration Results

Table 2 shows the calibration results of FBG center wavelength under different temperature environments. Figure 7 shows linear fitting results for FBG interfacial temperature of the specimen. The coefficient of determination for all grating temperature fitting lines exceeded 0.99, indicating a strong linear relationship between central wavelength and temperature across all sensors. The relationship between offset of central wavelength and temperature variation consistently follows:(4)$$\Delta \lambda =k_{T}\cdot \Delta T$$

Linear fitting of central grating wavelength against temperature yielded temperature coefficients (*K*_T_) and linear correlation coefficients of determination (R^2^) for interfacial gratings, as summarized in Table 3. In addition, the temperature at the beginning of the experiment was 22.6 degrees, the initial center wavelength (*λ*_0_) of FBGs are also provided in Table 3.

Variations in fiber-grating temperature coefficients result from implantation process issues and interactions between gratings and contact materials, such as epoxy resin core rods and silicone rubber housing. Although the sensitivity coefficients of each FBG vary, in all subsequent data analysis, the temperature reading for each FBG was calculated using its own unique *K*_T_ value using the Formula (1). This method has been widely demonstrated to effectively compensate for sensitivity changes induced by factors like packaging strain, thereby ensuring the accuracy of the temperature reading at each location.

Furthermore, the multi-point FBG system is based on Wavelength Division Multiplexing (WDM). Each FBG sensor is fabricated to have a distinct and stable central Bragg wavelength. As shown in Table 2, the adjacent FBG sensors are spaced with a spectral separation of about 10 nm, which is significantly larger than the typical wavelength shift (usually less than 1–2 nm) induced by temperature or strain variations in this application. This design fundamentally prevents the spectra of adjacent sensors from overlapping, thereby eliminating the risk of signal interference. In addition, the placement of the FBG sensors along the insulator was also strategically planned, to ensure that the FBG sensors, pre-embedded at the epoxy resin core rod–silicone rubber interface, do not cause mutual interference.

### 5.2. FBG Temperature and Visible Images During Flashover of Fiber-Optic Composite Insulator

#### 5.2.1. FBG Temperature

Since the FBG positions in optical fibers #1 and #4 are identical, the strong consistency in their interfacial temperature variations, as shown in Figure 8, effectively validates the reliability and accuracy of the temperature measurement method.

Figure 9 shows interfacial FBG temperature variations during AC salt fog flashover on the specimen. Fibers 1–4# were installed along the rod core between the insulator sheds to monitor corona and surface discharges. The interfacial temperature rise in FBGk3 (k = 1~4) was significantly higher than that of FBGk1 and FBGk2, revealing the most intense surface discharge activity on the sheath at the high-voltage side, a finding subsequently confirmed by visible-light imaging. The distinctive placement of fiber 5# directly beneath the sheds enabled the effective monitoring of surface and air-gap discharge along shed surfaces. A sharp temperature increase was observed at the FBG52 interface when flashover trip occurred at 316 s.

#### 5.2.2. Arc Visible-Light Images

Figure 10 shows visible-light images of arc discharge during AC salt fog flashover on the fiber-optic composite insulator. The upper end of insulator was grounded, while the lower end served as the positive electrode. Gratings 1, 2, and 3 were distributed from top to bottom. Voltage elevation was accompanied by corona or surface discharge. At flashover voltage, electric arcs formed a complete conducting path bridging both electrodes. Comparison with Figure 9 reveals consistency between arc locations and positions of FBG interfacial temperature rise, demonstrating the capability of fiber-grating composite insulators for online discharge monitoring. Additionally, an increase in corona discharge frequency was observed prior to flashover trip.

### 5.3. Early Warning for AC Salt Fog Flashover of Fiber-Optic Composite Insulator

#### 5.3.1. Flashover Early Warning Based on FBG Interfacial Temperature Rise

Temperature serves as a critical parameter in corona discharge, partial arcing, and flashover processes on hydrophobic surfaces of composite insulators. The fiber-optic composite insulator developed in this study enables monitoring of discharge processes and provides flashover early warning based on interfacial temperature rise rates. Each FBG monitors operational status at different insulator locations. Corona discharge occurs predominantly near high-voltage end fittings, resulting in the most pronounced interfacial temperature rise at Grating 3 among Fibers 1–4#. Fiber 5# is designated for flashover warning, with interfacial temperature rise rates during flashover listed in Table 4.

At 250 s into the discharge process, the interfacial temperature rise rate of FBG52 reached a maximum value of 4.69 × 10^−2^ °C/s. No surface arcing was observed on the insulation surface for the subsequent 60 s. Until another flashover occurred at 310 s, the interfacial temperature rise rate of FBG51 reached 4.88 × 10^−2^ °C/s, which was designated as the early warning threshold. A flashover trip was triggered at 316 s when the interfacial temperature rise rate of FBG53 attained 4.96 × 10^−2^ °C/s.

The flashover process of composite insulators under salt fog contamination conditions is inherently stochastic, characterized by factors such as non-uniform pollution layers, random discharge behavior, and uneven moisture/heat distribution. But a highly consistent critical temperature rise rate threshold (4.6 × 10^−2^ °C/s) was obtained by conducting flashover tests on silicone rubber plates of equal thickness. This indicates that the threshold may be closely related to the basic thermophysical properties of the insulating material and the critical discharge energy that causes flashover.

#### 5.3.2. Flashover Early Warning Based on Visible Images of Arc

AC salt fog flashover processes on fiber-optic composite insulators involve repeated discharge cycles of arc ignition, extinction, and re-ignition. To simulate the online monitoring of power transmission lines, a camera operating at 30 FPS was employed in this study, corresponding to a time interval of 33.33 ms between consecutive images, enabling effective capture of each discharge. Unlike high-speed camera recordings, this method acquires arc images throughout the entire discharge process. Three features were extracted: Blight area ratio (r), R-channel deviation (d), and mean saturation S. The characteristics of arc images at various discharge stages are shown in Figure 11a.

To avoid the bias introduced by subjective human assignment of weights, this study employs the entropy weight method to objectively assign weights to the aforementioned three components, thereby proposing an adaptive flashover threshold calculation method. This process is entirely data-driven, grounded in a rigorous mathematical theoretical foundation, which ensures the objectivity and scientific validity of the weight allocation results. The specific calculation workflow is as follows:

Three features were extracted from each image sample to form the original data matrix:(5)$$X=\left[\begin{array}{ccc} x_{11} & x_{12} & x_{13}\\ \begin{array}{l} x_{21} \end{array} & \begin{array}{l} x_{22} \end{array} & \begin{array}{l} x_{23} \end{array}\\ \vdots & \vdots & \vdots \\ x_{n1} & x_{n2} & x_{n3} \end{array}\right]\;\;\;$$
where *n* represents the number of samples and *x_ij_* denotes *j*-th feature value of *i*-th sample, the original matrix was normalized as follows:(6)$$p_{ij\;}=\frac{x_{ij}\;-\min(x_{j}\;)}{\max(x_{j}\;)-\min(x_{j}\;)}$$ Here, max(*x_j_*) and min(*x_j_*) are the maximum and minimum values of *j*-th feature across all samples, respectively. Each element in matrix *X* is then converted into its proportion, denoted as *p_ij_*, for the subsequent calculation of its information entropy e*_j_*:(7)$$r_{ij\;}=\frac{p_{ij\;}}{\sum_{i=1}^{n}p_{ij\;}};e_{j}=-\frac{\sum_{i=1}^{n}r_{ij}\ln(r_{ij})}{\ln(n)}$$ Where the constant ln(*n*) ensures that the information entropy falls within the interval [0, 1]. The differentiation coefficient is defined as:(8)$$g_{j\;}=1-e_{j\;}$$ The final weight *w_j_* is determined by the ratio of its differentiation coefficient to the sum of all differentiation coefficients:(9)$$w_{j}=\frac{g_{j}}{\sum_{k=1}^{3}g_{k}}$$

A discharge-region ratio *B* is defined as the weighted average of three arc image characteristics to achieve flashover early warning, as expressed as:(10)$$B=w_{1}\cdot r+w_{2}\cdot d+w_{3}\cdot S$$
where Equations (5)–(10) represent the original derivation process in this study.

The weights of the three feature parameters in this sample are 0.287, 0.112, and 0.601, respectively. The proportion of highlighted areas across various discharge stages is shown in Figure 11b. It is proposed that no discharge or only corona discharge occurs when the discharge-region ratio remains below 10%. Partial discharge arc development takes place within the 10–46.7% range. A ratio exceeding 46.7% indicates an imminent flashover trip, with final flashover trip occurring at 67.6%.

#### 5.3.3. Dual-Parameter Collaborative Flashover Early Warning

A key finding of our study is that not every arc leads immediately to flashover. A prime example is the transient arc observed around 250 s (e.g., at FBG52), which caused a temperature rise rate approaching our threshold but subsequently extinguished. A warning system triggered solely by a transient arc or a single temperature spike would have generated a false alarm here.

Otherwise, as sketched in Figure 10, the visible-light camera can only monitor the front surface of the insulator. Arcs developing on the sides or back surface are completely invisible to the camera, leading to a high risk of missed alarms.

The method based on FBG and visible-light images is based on the fusion of different physical parameters, effectively avoiding false negatives and misjudgment caused by accidental factors of a single parameter and significantly improving the reliability of the warning method. Specifically, the flowchart of the flashover warning method is shown in Figure 12.

The early warning method is designed to operate in a dual-mode manner, which is critical for its computational feasibility. The FBG-based temperature monitoring serves as the primary, continuous, and low-computational-cost trigger. The image feature extraction is activated only when the FBG system detects abnormal temperature rise rate of any grating. The image-processing algorithm are evaluated on an industrial computing platform with an AMD 5600× CPU and NVIDIA GeForce RTX 3080 graphics card (NVIDIA Corporation, Santa Clara, CA, USA). The data were analyzed using Python (version 3.8). The key quantitative metrics are as follows: The average time to process a single image frame (including feature extraction) is 11.34 ms, the image interval in this approach is 33.33 ms. Therefore, the proposed method is computationally feasible for real-time operational constraints.

## 6. Conclusions

This paper proposes a method for detecting interfacial temperature based on fiber-optic composite insulators, with a linear relationship between wavelength shift and temperature demonstrated through fiber-grating calibration experiments. A fiber-optic composite insulator embedded with five optical fibers, which contain three gratings in each fiber, was designed and fabricated. Gratings on optical fibers 1–4# are suitable for monitoring air-gap and corona discharge, while those on optical fiber 5# are adapted for discharge monitoring of surface and air-gap discharge along shed surfaces. AC flashover tests were conducted on the insulator in an artificial fog chamber simulating coastal salt fog environments. A flashover early warning method based on FBG interfacial temperature rise rate and visible discharge images are proposed: flashover trip is imminent when the temperature rise rate reaches 4.88 × 10^−2^ °C/s; a discharge-region ratio characteristic extracted from visible-light images provides an early warning at a threshold of 46.7%. The dual-parameter flashover early warning system proposed in this study, which simultaneously monitors visible-light images and interfacial temperature, exhibits enhanced robustness and a low false-positive rate. Experimental results indicate consistency between interfacial temperature rise locations in FBGs and arc positions in visible discharge images, confirming the effectiveness of the proposed method for monitoring flashover discharge. In future work, we will conduct more systematic research on different Equivalent Salt Deposit Density (ESDD) levels and different insulator geometries. This research offers a new approach for discharge monitoring and flashover warning of external insulation under AC salt fog conditions.

## Figures and Tables

**Figure 1 micromachines-16-01171-f001:**
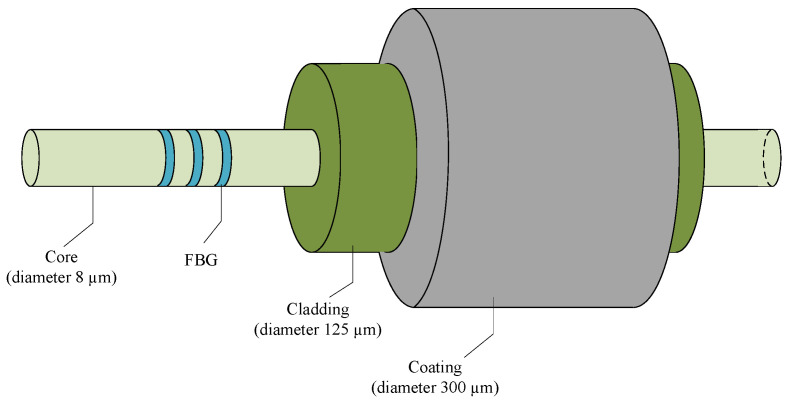
Schematic diagram of FBG structure.

**Figure 2 micromachines-16-01171-f002:**
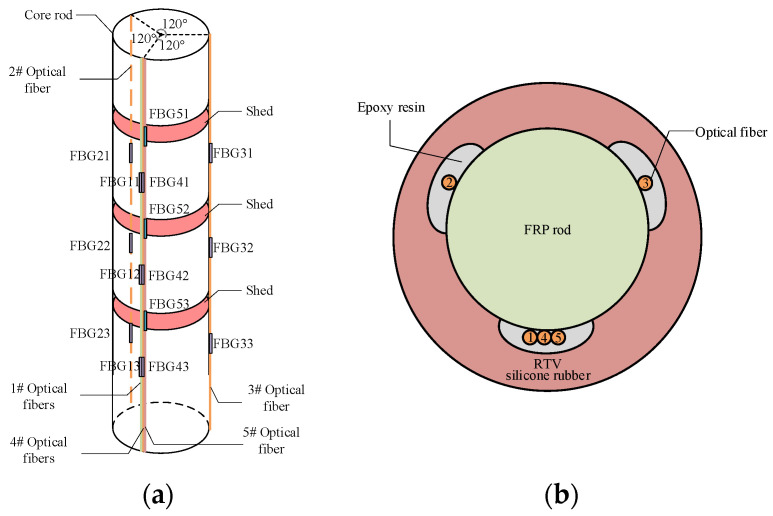
Stereoscopic and sectional schematic diagram of fiber optic composite insulator: (**a**) position distribution of FBGs; (**b**) sectional structure of insulator embedded with FBGs.

**Figure 3 micromachines-16-01171-f003:**
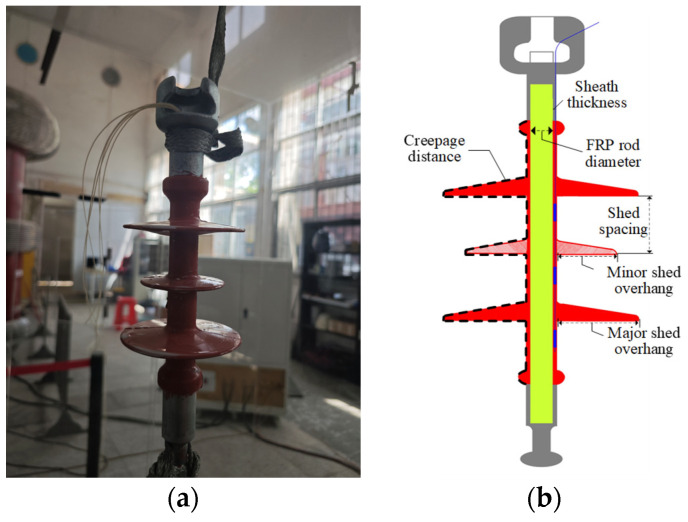
Fabricated fiber-optic composite insulator sample and its structural schematic diagram: (**a**) sample; (**b**) structural schematic diagram.

**Figure 4 micromachines-16-01171-f004:**
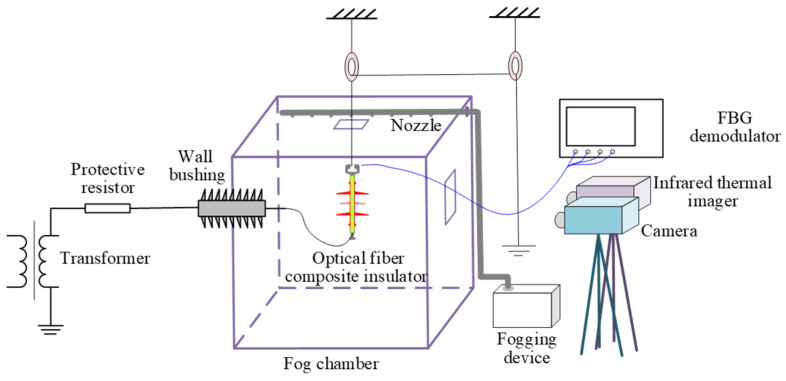
Flashover test system for fiber-optic composite insulators under salt fog conditions.

**Figure 5 micromachines-16-01171-f005:**
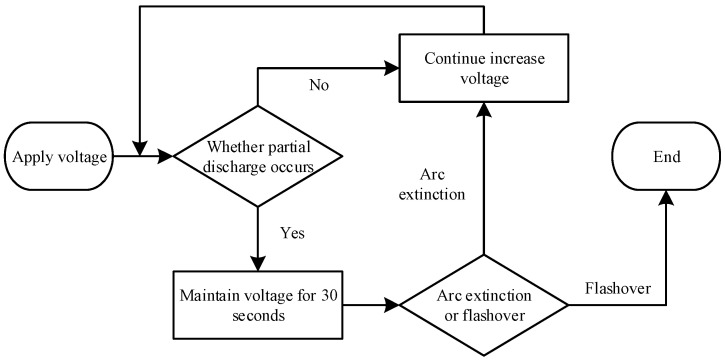
Flowchart of applied voltage method.

**Figure 6 micromachines-16-01171-f006:**
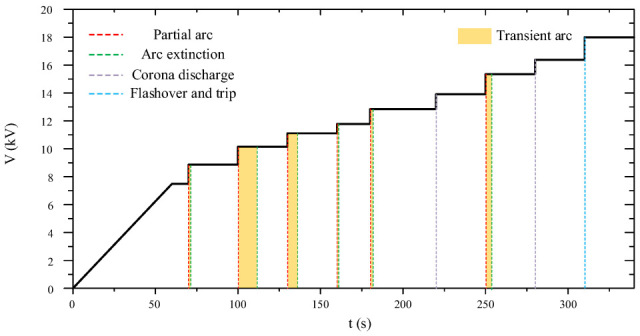
Applied voltage for AC salt fog flashover tests on the fiber-optic composite insulator.

**Figure 7 micromachines-16-01171-f007:**
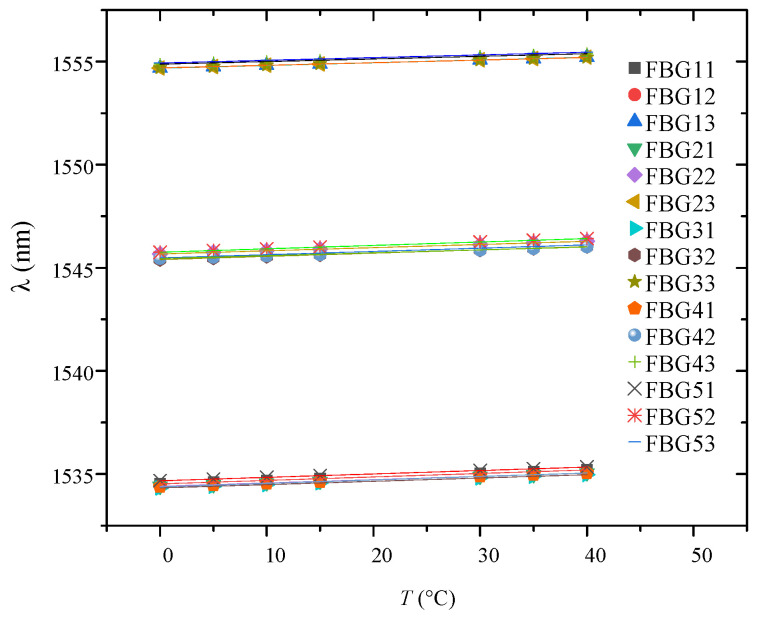
Linear fitting results of interface FBG temperature.

**Figure 8 micromachines-16-01171-f008:**
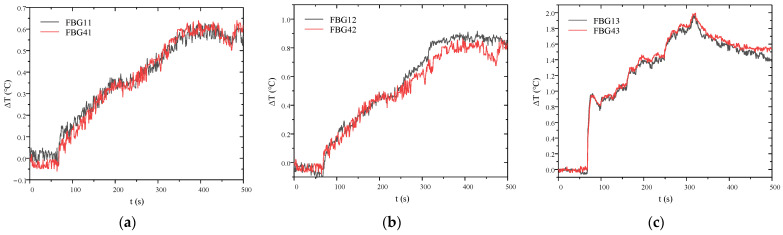
Comparison of temperature variation at the FBG interfaces between optical fiber #1 and #4. (**a**) FBG11 and FBG41; (**b**) FBG12 and FBG42; (**c**) FBG13 and FBG43.

**Figure 9 micromachines-16-01171-f009:**
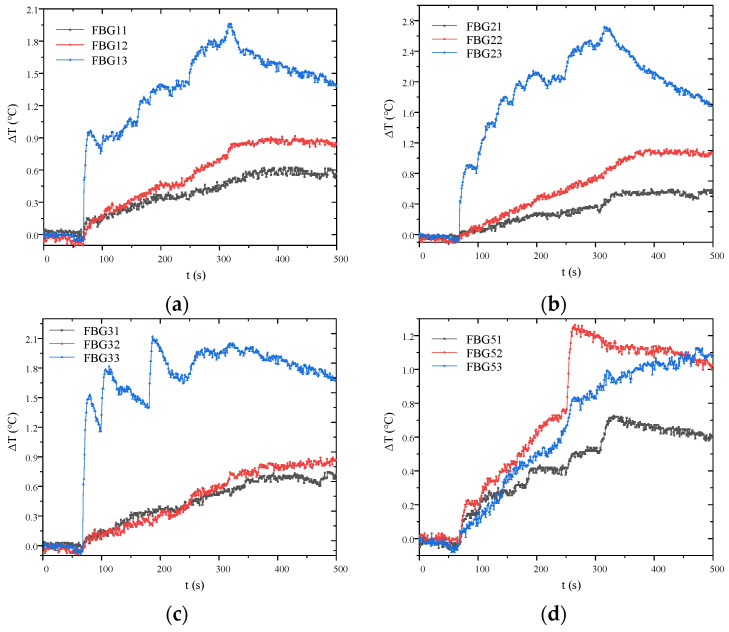
Interfacial temperature variations in FBGs during discharge of the fiber-optic composite insulator. (**a**) Fiber #1; (**b**) fiber #2; (**c**) fiber #3; (**d**) fiber #5.

**Figure 10 micromachines-16-01171-f010:**
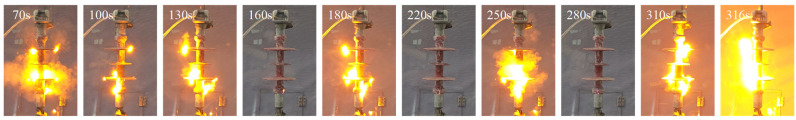
Visible-light images of arc discharge during flashover on the surface of fiber-optic composite insulator.

**Figure 11 micromachines-16-01171-f011:**
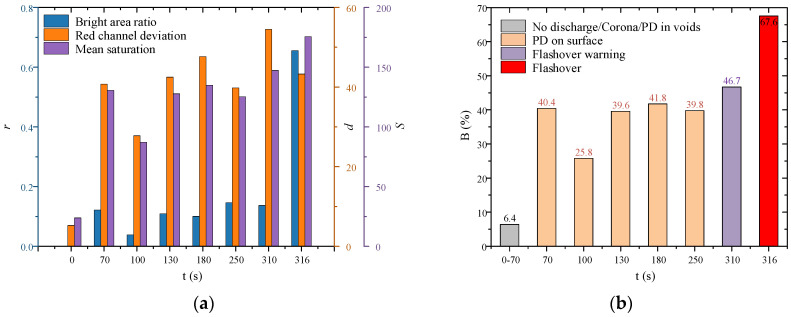
Characteristics of arc visible images during AC salt fog flashover in the fiber-optic composite insulator: (**a**) Blight area ratio, R-channel deviation, and mean saturation; (**b**) discharge-region ratio.

**Figure 12 micromachines-16-01171-f012:**
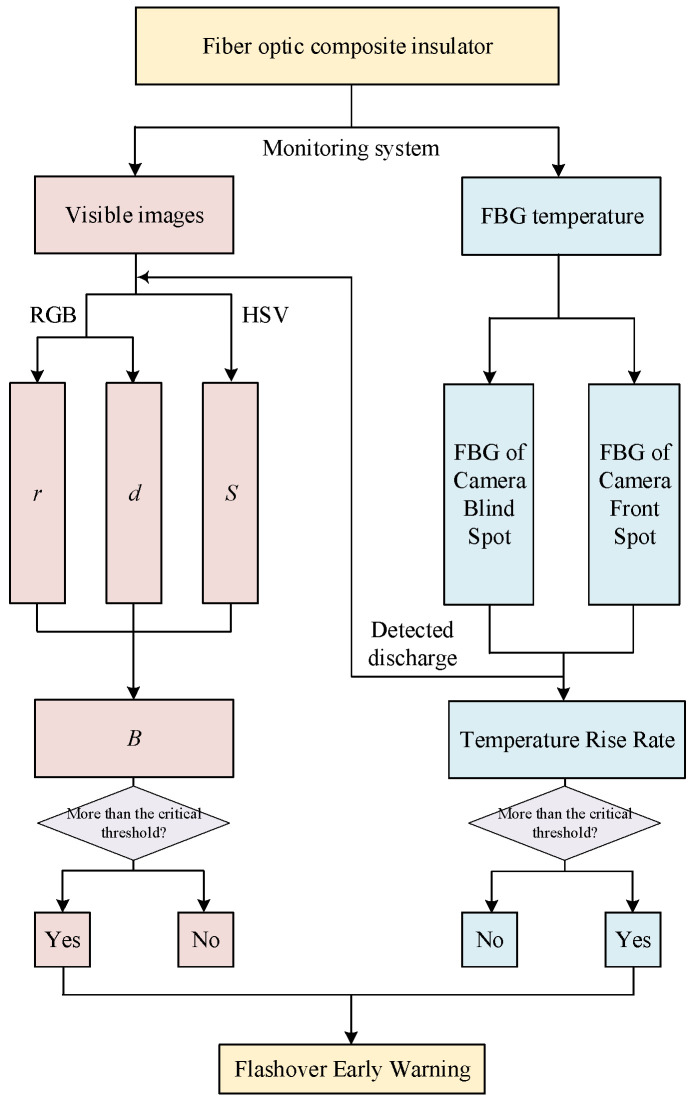
Flowchart of the dual-parameter collaborative flashover early warning method.

**Table 1 micromachines-16-01171-t001:** Parameters of the fiber-optic composite insulator.

Creepage Distance (mm)	Sheath Thickness (mm)	FRP Rod Diameter (mm)	Shed Spacing (mm)	Shed Overhang (mm)	
				Major Shed	Minor Shed
382	3.2	18	45	37.8	22.8

**Table 2 micromachines-16-01171-t002:** Calibration results of FBG center wavelength under various temperature environments.

Gratings	*λ* (nm)						
	0 °C	5 °C	10 °C	15 °C	30 °C	35 °C	40 °C
FBG11	1534.5273	1534.6102	1534.6932	1534.7761	1535.0365	1535.1177	1535.2012
FBG12	1545.4765	1545.5556	1545.6347	1545.7138	1545.9523	1546.0295	1546.1097
FBG13	1554.6813	1554.7469	1554.8126	1554.8782	1555.0771	1555.1396	1555.2072
FBG21	1534.4093	1534.4908	1534.5722	1534.6537	1534.8885	1534.9694	1535.0541
FBG22	1545.6679	1545.7469	1545.8258	1545.9048	1546.1223	1546.2042	1546.2896
FBG23	1554.6867	1554.7527	1554.8188	1554.8848	1555.0775	1555.1439	1555.2116
FBG31	1534.3274	1534.4093	1534.4911	1534.573	1534.8119	1534.8899	1534.973
FBG32	1545.397	1545.4766	1545.5562	1545.6358	1545.864	1545.9439	1546.0261
FBG33	1554.6934	1554.7578	1554.8223	1554.8867	1555.0723	1555.1344	1555.2024
FBG41	1534.3819	1534.4647	1534.5475	1534.6303	1534.8889	1534.9708	1535.0545
FBG42	1545.4586	1545.52	1545.5814	1545.6428	1545.8586	1545.9374	1546.0206
FBG43	1554.8739	1554.9379	1555.0019	1555.0659	1555.2595	1555.3188	1555.3825
FBG51	1534.6785	1534.7605	1534.8425	1534.9245	1535.1765	1535.2574	1535.3421
FBG52	1545.7533	1545.8364	1545.9195	1546.0026	1546.2494	1546.3313	1546.4171
FBG53	1554.9293	1554.9964	1555.0636	1555.1307	1555.3227	1555.3888	1555.4604

**Table 3 micromachines-16-01171-t003:** Temperature coefficients *K*_T,_ coefficients of determination R^2^, and initial center wavelength *λ*_0_ for FBGs.

Gratings	*K_T_* (pm/°C)	R^2^	*λ*_0_ at 22.6 °C (nm)
FBG11	16.59	0.9986	1534.9022
FBG12	15.82	0.9999	1545.834
FBG13	13.13	0.9999	1554.978
FBG21	16.29	0.9985	1534.7775
FBG22	15.79	0.9943	1546.0248
FBG23	13.21	0.9995	1554.9852
FBG31	16.37	0.9989	1534.6974
FBG32	15.92	0.9986	1545.7568
FBG33	12.89	0.9993	1554.9847
FBG41	16.56	0.9987	1534.7562
FBG42	16.27	0.9998	1545.7361
FBG43	12.80	0.9983	1555.1632
FBG51	16.40	0.9995	1535.0491
FBG52	16.62	0.9999	1546.1289
FBG53	13.43	0.9992	1555.2328

**Table 4 micromachines-16-01171-t004:** Interfacial temperature rise rate *β* of FBG5 during discharge process.

Gratings	*β* (°C/s)						
	70 s	100 s	130 s	180 s	250 s	310 s	316 s
FBG51	2.26 × 10^−2^	1.83 × 10^−2^	3.35 × 10^−2^	2.44 × 10^−2^	1.22 × 10^−2^	4.88 × 10^−2^	3.66 × 10^−2^
FBG52	2.41 × 10^−2^	1.81 × 10^−2^	2.41 × 10^−2^	2.41 × 10^−2^	4.69 × 10^−2^	×	1.81 × 10^−2^
FBG53	2.79 × 10^−2^	2.73 × 10^−2^	2.23 × 10^−2^	×	1.19 × 10^−2^	×	4.96 × 10^−2^

## Data Availability

The data presented in this study are openly available in [Metadata] at [https://doi.org/10.5281/zenodo.17088481].

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
