# Peer review of "Early Warning of AC Salt Fog Flashover on Composite Insulators Using Fiber Bragg Grating Sensing and Visible Arc Images"

_micromachines, 2025, doi:10.3390/mi16101171_

Round 1
Reviewer 1 Report
Comments and Suggestions for Authors
This study presents an FBG-based monitoring method for detecting flashovers in coastal power grid insulators. The authors stated a correlation between interfacial temperature and FBG wavelength was established, enabling early warning with high accuracy under salt fog conditions using temperature and visual discharge features. However, the current manuscript lacks novelty. Some critical concerns have remained.
- Are the FBGs used in this study commercially sourced or fabricated by the authors? This requires clarification. Throughout the manuscript, no original FBG reflection spectra are provided as references. These details are missing and need to be clearly addressed.
- The calibration in section 4.1 needs further clarification. The FBG spectra at various temperature should be provided.
- Given that the FBGs are embedded beneath a layer of RTV silicone rubber, how does the thermal resistance or heat transfer delay introduced by the RTV layer affect the real-time temperature accuracy and response speed of the FBG sensor during fast-rising discharge events? Has this delay been quantified or modeled?
- The bonding process between the FBGs and the FRP rod uses epoxy adhesive, and the composite structure is exposed to high humidity and salinity. Has the long-term stability of FBG adhesion been tested under cyclic thermal and environmental loading? Could sensor detachment or drift occur over prolonged field usage? Need further information.
- Table 2 shows notable variation in temperature sensitivity coefficients (KT ranges from 12.8 to 16.6 pm/°C). What are the possible causes of this variation — implantation angle, adhesive thickness, or strain from encapsulation? Should position-dependent calibration be implemented to ensure accurate temperature readings across all grating locations?
- The arc discharge early warning system uses a 30 FPS camera, which may miss fast transient events or underestimate arc propagation speed. Is this temporal resolution sufficient to capture rapid arc growth dynamics?
- The flashover early warning threshold is based on a fixed temperature rise rate (4.88×10⁻² °C/s). Has this threshold been validated across multiple environmental conditions (humidity, fog density, salt concentration)? And how sensitive is the system to false positives or delayed triggering under fluctuating ambient or surface conditions?
Author Response
Comment 1: Are the FBGs used in this study commercially sourced or fabricated by the authors? This requires clarification. Throughout the manuscript, no original FBG reflection spectra are provided as references. These details are missing and need to be clearly addressed.
Response 1:
We sincerely thank the reviewer for raising this important point regarding the origin and characterization of the FBGs.
The FBGs themselves were commercially sourced to ensure consistency and reliability in their optical performance. However, the process of embedding these FBGs into the composite insulator—resulting in the optical fiber composite insulator specifically designed for this study—was entirely conducted by our research team. The detailed fabrication process, including the specific procedures and technical steps involved, is thoroughly described in Section. 3 of the manuscript. This section also specifies the spatial layout and precise positioning of all the FBGs within the insulator structure.
In response, we have now included the initial central wavelengths of all the FBGs at 22.6 ℃ (temperature at the beginning of experiment) in Table 3 located in Section 5.1. We are grateful for the reviewer’s suggestion concerning the original FBG reflection spectra.
Comment 2: The calibration in section 4.1 needs further clarification. The FBG spectra at various temperature should be provided.
Response 2:
We thank the reviewer for this valuable suggestion to improve the clarity of our calibration process. In direct response to the comment, we have provided the complete dataset of wavelength shifts for all 15 FBGs across the seven temperature sets as Supplementary Table 3. Additionally, we have updated Figure 7 in the manuscript to better illustrate the calibration curves for representative sensors.
The central premise of our calibration approach is that the temperature sensitivity coefficient (KT) for each fiber is constant and well-characterized. Therefore, the absolute initial wavelength at time zero is less critical than the relative wavelength shift (Dl) throughout the experiment. All subsequent temperature changes reported in the study are derived from this relative shift using the formula ΔT=Δλ/KT. This method ensures that the trends and conclusions regarding temperature evolution in the subsequent content are accurate and reliable.
We believe these revisions have significantly strengthened the manuscript, and we are grateful to the reviewer for prompting this improvement in rigor.
Comment 3: Given that the FBGs are embedded beneath a layer of RTV silicone rubber, how does the thermal resistance or heat transfer delay introduced by the RTV layer affect the real-time temperature accuracy and response speed of the FBG sensor during fast-rising discharge events? Has this delay been quantified or modeled?
Response 3:
We sincerely thank the reviewer for raising this critical point regarding the dynamic response performance of our FBG sensors, which is indeed a vital factor for accurate thermal measurement. We fully agree that any packaging material introduces thermal inertia, which theoretically can cause a measurement delay.
In our study, the RTV silicone rubber layer, with a nominal thickness of 3.2 mm, was primarily applied to provide essential electrical insulation and mechanical protection for the optical fiber, rather than serving as a thermal management coating. And also this thickness also meets the standards for 10 kV composite insulators used in grid operation.
It is correct that the time constant for the entire RTV layer to reach thermal equilibrium is on the order of hundreds of seconds. However, this does not preclude the detection of rapid temperature changes during a high-energy discharge event. The explanation lies in the immense power density of the local arc. In this research:
The pre-flashover arc acts as an intense, localized heat source (with temperatures exceeding 5000~10000 K), injecting a large amount of thermal energy into the system rapidly; The FBG measures the temperature at its specific location in real-time. While the entire RTV layer may not reach equilibrium quickly, the powerful heat flux from the arc causes a rapid and significant temperature rise at the FBG site well before full equilibrium is achieved. The substantial temperature rise rates we observed (e.g., 0.046 °C/s) are direct evidence of this strong heat flux propagating through the medium. This is analogous to heating a thick metal block with a welding torch: the entire metal block's response is slow, but the point of application can become red-hot very quickly due to the high input power.
In summary, the thermal inertia of the RTV layer defines the speed of heat diffusion, but the intense energy of the discharge event provides the heat flux that creates a measurable and rapid temperature response at the sensor. Therefore, the observed rapid interface temperature rise is entirely consistent with the physics of heat transfer and robustly indicates the critical discharge activity preceding flashover.
Comment 4: The bonding process between the FBGs and the FRP rod uses epoxy adhesive, and the composite structure is exposed to high humidity and salinity. Has the long-term stability of FBG adhesion been tested under cyclic thermal and environmental loading? Could sensor detachment or drift occur over prolonged field usage? Need further information.
Response 4:
We sincerely thank the reviewer for raising this critically important question regarding the long-term reliability of the sensor embedding technique, which is essential for the practical application of our monitoring method.
While the primary focus of this study was on the dynamic thermal response of the insulator during discharge events, and thus dedicated accelerated aging tests on the samples were not conducted within the scope of this work, we would like to address this concern from the following two perspectives:
Evidence from existing literature and grid operation: As mentioned in the introduction of our manuscript, the technology of embedding FBGs into composite insulators to create "optical fiber composite insulators" for online monitoring has been developed and deployed for years [17]. These field cases demonstrate that with an optimized bonding technique, the sensors can maintain mechanical integrity and optical stability under long-term outdoor exposure, including high humidity, salinity, and UV radiation. The bonding process adopted in our study is based on these proven methodologies.
Reinforcement measures implemented in this study:
1) Rigorous Surface Preparation: The FRP rod surface was meticulously sanded, cleaned, and chemically treated prior to bonding to maximize adhesion strength.
2) Selection of High-Performance Epoxy: We have prioritized the use of epoxy resin with low shrinkage rate, high toughness, and excellent resistance to hygrothermal aging. This selection strategy was aimed at effectively mitigating the internal stresses arising from the coefficient of thermal expansion mismatch among the FRP, epoxy, and the optical fiber. The high toughness of the epoxy allows it to absorb stresses through micro-deformation, thereby protecting the brittle optical fiber and maintaining the long-term integrity of the bonding interface.
3) Structural Protection Design: The FBG sensors are ultimately encapsulated and protected by the insulator's sheds. This structure provides excellent mechanical protection and environmental isolation for the bonding points, significantly reducing the risk of direct mechanical impact or medium erosion.
In conclusion, based on the validated field experience and the optimized process employed in our experiment, the probability of sensor detachment or significant drift caused by bonding failure of composite insulators is very low. Meanwhile, we have added instructions on selecting high-performance epoxy resin in the Section 3. We fully agree with the reviewer that subjecting the sensor to accelerated aging tests under more severe cyclic loading is a crucial step towards future productization, and this suggestion provides valuable direction for our subsequent research.
Comment 5: Table 2 shows notable variation in temperature sensitivity coefficients (KT ranges from 12.8 to 16.6 pm/°C). What are the possible causes of this variation — implantation angle, adhesive thickness, or strain from encapsulation? Should position-dependent calibration be implemented to ensure accurate temperature readings across all grating locations?
Response 5:
We sincerely thank the reviewer for this insightful observation and the critical questions raised. The variation in KT values is indeed a key phenomenon in packaged FBG sensors, and the potential causes mentioned are highly relevant and accurate. Our analysis is as follows:
1) Analysis of the causes for variation: The residual strain induced by the encapsulation process, as suggested by the reviewer, is the primary cause of the KT variation. Despite our efforts to maintain process consistency, the complex procedure of embedding FBGs into the composite insulator inevitably introduces minor and non-uniform residual stresses. This is mainly due to the mismatch in the coefficients of thermal expansion (CTE) among the FRP rod, epoxy adhesive, and silicone rubber housing materials, which generates stress during the curing and cooling process. Minor variations in the implantation angle and adhesive thickness exacerbate this local non-uniformity in stress distribution. These factors collectively cause the KT values of different FBGs to deviate from the theoretical value for bare fiber (~10 pm/°C).
2) About the necessity and implementation of position-dependent calibration: The reviewer raises a critical point. We wish to emphasize that a certain degree of variation in KT values is normal and acceptable for packaged FBG sensors. The key to ensuring measurement accuracy lies not in pursuing absolute consistency of KT values across all gratings, but in performing an independent and precise calibration for each individual sensor.
Therefore, we fully endorse the reviewer's suggestion, and the core calibration procedure in this study strictly adhered to this principle of "position-dependent calibration." The distinct KT values presented in Table 2 are the direct outcome of this process. In all subsequent data analysis, the temperature reading for each FBG was calculated using its own unique KT value using the formula Equation (1). This method has been widely demonstrated to effectively compensate for sensitivity changes induced by factors like packaging strain, thereby ensuring the accuracy of the temperature reading at each location, even though their sensitivity coefficients themselves vary. In essence, our focus is on the accuracy of each sensor's output, not the uniformity of its sensitivity. The current calibration methodology is sufficient to guarantee the reliability of our data.
In additional, we have added a detailed explanation of this issue at the end of Section 5.1.
We appreciate the reviewer's questions, which have allowed us to clarify the calibration procedure more explicitly, underscoring our commitment to data accuracy.
Comment 6: The arc discharge early warning system uses a 30 FPS camera, which may miss fast transient events or underestimate arc propagation speed. Is this temporal resolution sufficient to capture rapid arc growth dynamics?
Response 6:
We sincerely thank the reviewer for this important question. We fully understand the concern regarding temporal resolution and appreciate the opportunity to provide further clarification.
Our preliminary studies using high-speed cameras (frame rate: 4000 FPS) have confirmed that individual AC partial discharge events on composite insulator surfaces under salt fog conditions typically have a duration of 60 milliseconds or longer. This finding is crucial for assessing the suitability of the 30 FPS camera.
Based on this, we believe that the 30 FPS frame rate is an adequate and well-considered choice for our early warning system for the following reasons:
1) The frame rate covers the discharge duration: The sampling interval of 30 FPS is 33.3 ms. Since the duration of a single discharge event (≥60 ms) is significantly longer than this interval, at least 2 consecutive frames are guaranteed to capture the event. The camera's principle is that any discharge activity occurring during the 33.3 ms integration (exposure) time will be recorded. Therefore, the system does not miss discharges of significant energy and can effectively track their evolution of next discharge event.
2) Meeting the need for long-term monitoring: The core of this research is to reveal the complete energy accumulation regularity over hundreds of seconds—from the initial arc inception to final flashover—and to provide warnings accordingly. High-speed cameras, limited by storage capacity, cannot achieve such prolonged, uninterrupted recording. The 30 FPS industrial camera perfectly solves the challenge of long-duration, continuous recording, which is the foundation for reliable early warning.
3) Balance between early warning criteria and engineering cost: The warning criteria of our system are based on the continuous trends of arc characteristics (e.g., intensity, area) constructed on a timescale of seconds. The sequence provided by 30 FPS is sufficient to accurately depict these trends. Pursuing a higher frame rate offers diminishing returns for our warning objective while significantly increasing system cost and complexity. Thus, 30 FPS represents the optimal trade-off between ensuring effectiveness and maintaining engineering practicality.
In additional, we have added a detailed explanation of this issue in Section 4.1.
We thank the reviewer again for the comment, which has allowed us to better articulate the engineering rationale and physical basis of our study.
Comment 7: The flashover early warning threshold is based on a fixed temperature rise rate (4.88×10⁻² °C/s). Has this threshold been validated across multiple environmental conditions (humidity, fog density, salt concentration)? And how sensitive is the system to false positives or delayed triggering under fluctuating ambient or surface conditions?
Response 7:
We sincerely thank the reviewer for these two critical questions, which address the core issues of transitioning our early warning method from the laboratory to practical field applications.
1) Validation across environmental conditions:
We fully agree that validating the threshold’s stability under a wider range of environmental conditions is a crucial next step for applied research. At this stage of our study, the primary aim was to propose and validate the feasibility of the fundamental principle—using the interface temperature rise rate as a flashover precursor. Therefore, all experiments were conducted under carefully controlled, specific conditions simulating a typical coastal environment to establish a clear “proof of concept”.
However, we have preliminary evidence suggesting an inherent robustness of the criterion. As mentioned, an independent experiment on a silicone rubber flat plate yielded a highly consistent critical temperature rise rate threshold (0.046 °C/s). This indicates that the threshold may be closely tied to the fundamental thermophysical properties of the insulating material and the critical discharge energy leading to flashover, rather than being overly sensitive to a particular set of environmental parameters. This provides an optimistic indication of its potential stability.
2) Sensitivity to false positives and delayed triggering:
This is central to the design of any warning system. Our system incorporates the following mechanisms to enhance robustness against these issues: Mitigating False Positives: The warning criterion does not rely solely on the temperature rise rate in isolation. It requires temporal correlation with a pre-flashover or sustained arc state observed in the visible light images. This fusion of multi-source information (temperature + optics) effectively filters out non-critical temperature rises caused by ambient fluctuations or minor local discharges, thereby significantly reducing the false positive rate.
3) Avoiding Delayed Triggering:
The threshold is designed to capture an instantaneous physical signal indicating that the arc energy accumulation has reached a critical state. Once this threshold is exceeded, flashover typically occurs within seconds. The triggering is nearly instantaneous as long as the sensor is functional. The dual-parameter flashover early warning system proposed in this study, which simultaneously monitors visible light images and inter-facial temperature, exhibits enhanced robustness and a low false positive rate.
In additional, we have added a detailed explanation of this issue in Section 5.3.1 and Section 6.
In summary, we acknowledge that further validation under a broader range of conditions is a valuable direction for future work. Nevertheless, the consistent cross-sample correlations and the multi-sensor monitoring results observed in this study remain valid. They provide a solid foundation and clear guidance for optimizing practical applications.

Reviewer 2 Report
Comments and Suggestions for Authors
This paper presents a composite insulator AC salt-fog flashover early warning method integrating FBG temperature sensing and visible arc image analysis. The authors fabricate a fiber-composite insulator with embedded FBG sensors, validated through artificial salt-fog chamber tests. The proposed method combines temperature rise rate thresholds derived from FBG wavelength shifts and image features for multi-parameter early warning. While the approach shows promise in laboratory settings, critical technical limitations persist. Overall, it is an interesting work, and can be considered for publication after revision. Please follow my comments below before re-submission.
-Have the authors evaluated how varying salt deposition densities influence FBG temperature rise rate thresholds across different insulator geometries?
-The experimental design lacks comparative analysis with existing single-parameter warning methods, limiting claims of superiority.
-What quantitative metrics demonstrate the computational feasibility of the proposed image feature extraction under real-time operational constraints?
-The RTV-FBG interface’s long-term durability under thermal cycling and UV exposure requires accelerated aging tests beyond laboratory conditions.
-Could the fixed threshold-based warning system adapt to dynamic environmental parameters like ambient humidity or wind speed variations?
-The limited sample size undermines statistical robustness of flashover characteristics across manufacturer variations.
-The literature review demonstrates adequate engagement with recent advancements in early warning. More publications on application of distributed fibre optic sensors (DOI: 10.1080/10298436.2025.2479645) should be incorporated to strengthen context.
-How do the authors reconcile discrepancies between laboratory salt-fog conditions and actual field pollution patterns affecting insulator performance?
-The absence of adaptive algorithms for threshold calibration based on real-time environmental feedback constitutes a methodological gap.
-Have the authors verified the proposed method’s efficacy across diverse geographical altitudes where atmospheric pressure impacts arc behavior?
-The paper fails to address potential signal interference between adjacent FBG sensors in multi-point monitoring configurations.
Please double check english expressions.
Author Response
Comment 1: Have the authors evaluated how varying salt deposition densities influence FBG temperature rise rate thresholds across different insulator geometries?
Response 1:
We sincerely thank the reviewer for this insightful and important question regarding the impact of equivalent salt deposit density (ESDD) and different insulator geometries on the FBG temperature rise rate threshold.
1) The primary objective of this initial study was to validate the fundamental feasibility of using FBG sensors to detect the pre-flashover condition and determine a characteristic temperature rise rate threshold. Consequently, our experiments were designed to simulate a typical coastal saline environment using a fixed, representative ESDD value. The research focused specifically on 10kV composite insulators, with supplementary validation on flat silicone rubber sheets. We acknowledge that a systematic investigation into the influence of varying ESDD levels and different insulator geometries was not within the immediate scope of this paper. We have added a statement in the revised manuscript (Section 5) to clarify this limitation.
2) Although ESDD undoubtedly affects the starting voltage of arcing and flashover, our experiments conducted on silicone rubber plates have shown that the critical flashover temperature rise rate threshold is also 0.046 and 0.048 °C/s when flashover occurs at a voltage of 6.96kV, indicating that this threshold is mainly a manifestation of the thermal physical properties of silicone rubber. This provides a certain degree of robustness for the recognition threshold of a given material system. We have provided additional explanations on this point in Section 5.3.1.
3) The reviewer's suggestion is highly valuable. A comprehensive study examining the effects of ESDD, non-soluble deposit density (NSDD), and insulator geometry on the FBG threshold is indeed the essential next step for translating this laboratory-based technique into a field-applicable warning system. We will conduct further analysis on insulators of different voltage levels as a core component of our subsequent research.
We thank the reviewer again for this constructive comment.
Comment 2: The experimental design lacks comparative analysis with existing single-parameter warning methods, limiting claims of superiority.
Response 2:
We thank the reviewer for this critical comment, which helps us clarify the key advantage of our multi-parameter approach over conventional methods. The core innovation and superiority of our method lie in its fundamental principle of multi-parameter fusion to overcome the inherent limitations of single-parameter warnings. Based on our results, we have revised the manuscript to more cautiously and clearly articulate this advantage by explaining the following points:
1) False positives of FBG temperature rise rate only: A key finding of our study is that not every arc leads immediately to flashover. A prime example is the transient arc observed around 250s (e.g., at FBG52), which caused a temperature rise rate approaching our threshold but subsequently extinguished. A warning system triggered solely by transient arc or a single temperature spike would have generated a false alarm here.
2) False negatives with visible imaging only: As the reviewer can see in our visible image data (and as sketched in Figure. 10), the visible-light camera can only monitor the front surface of the insulator. Arcs developing on the sides or back surface are completely invisible to the camera, leading to a high risk of missed alarms.
3) Superiority through synergy: The method based on FBG and visible light images is based on the fusion of different physical parameters, effectively avoiding false negatives and misjudgment caused by accidental factors of a single parameter and significantly improving the reliability of the warning method.
We have added explanations in Section 5.3.3 regarding the single parameter issue mentioned above in the corresponding section of the article. We would like to thank the reviewers for their suggestions, which have helped us better demonstrate the advantages of dual parameter collaboration in this paper.
Comment 3: What quantitative metrics demonstrate the computational feasibility of the proposed image feature extraction under real-time operational constraints?
Response 3:
We sincerely thank the reviewer for raising this crucial point regarding the real-time applicability of our method. The following explanation are provided regarding this issue:
1) Our proposed early warning system is designed to operate in a dual-mode manner, which is critical for its computational feasibility. The FBG-based temperature monitoring serves as the primary, continuous, and low-computational-cost trigger. It runs in real-time with a sampling rate of 10 HZ, consuming minimal computational resources.
2) The image feature extraction is not required to run continuously at high speed. It is activated only when the FBG system detects abnormal temperature rise rate of any grating. This design significantly reduces the average computational load.
3) To directly address the reviewer's question, we evaluated the image processing algorithm on an industrial computing platform with an AMD 5600X CPU and NVIDIA 3080 graphics card. The key quantitative metrics are as follows: The average time to process a single image frame (including feature extraction) is 11.34ms, the image interval in this approach is 33.33ms. Therefore, the proposed method is computationally feasible for real-time operational constraints.
We have added these quantitative performance data and a discussion on the dual-mode operation concept to the revised manuscript in Section 5.3.3 to strengthen our claim. We thank the reviewer again for prompting us to include this important validation.
Comment 4: The RTV-FBG interface’s long-term durability under thermal cycling and UV exposure requires accelerated aging tests beyond laboratory conditions.
Response 4:
We greatly appreciate the reviewer's insightful comment regarding the long-term durability of the RTV-coated FBG sensor. We completely agree that assessing the performance under accelerated aging conditions, such as thermal cycling and UV exposure, is a crucial step for validating the sensor's suitability for long-term outdoor deployment. Please allow us to clarify the current scope of our study and potential future work content.
The central objective of this manuscript is to establish the fundamental principle and short-term efficacy of using the RTV-FBG interface for flashover early-warning. Our experimental results successfully demonstrate its high sensitivity and reliability in capturing the critical pre-flashover thermal signature under simulated environmental stresses (e.g., salt fog). We acknowledge that a comprehensive long-term aging study was beyond the scope of this initial proof-of-concept investigation.
Although full-scale accelerated aging tests were not conducted herein, the materials and design were chosen with long-term durability in mind. The RTV coating itself is a commercially available material specifically formulated for outdoor high-voltage insulation, possessing inherent resistance to UV radiation and thermal degradation. The FBG sensor is fabricated in silica glass, which is highly stable under the expected temperature ranges.
As mentioned in the introduction of our manuscript, the technology of embedding FBGs into composite insulators to create fiber-optical composite insulators for online monitoring has been developed and deployed for years. These field cases demonstrate that the sensors can maintain mechanical integrity and stability under long-term outdoor exposure, including high humidity, salinity, and UV radiation. In addition, we added some explanations in Section 3. We fully agree with the reviewer that addressing the long-term durability is indeed the essential next step towards the practical implementation of this technology, and this suggestion provides valuable direction for our subsequent research.
Comment 5: Could the fixed threshold-based warning system adapt to dynamic environmental parameters like ambient humidity or wind speed variations?
Response 5:
We sincerely thank the reviewer for this exceptionally insightful question. It is a significant challenge for deploying any fixed-threshold warning system in a highly variable outdoor environment. The fixed threshold proposed in this study, while effective under the controlled laboratory conditions of our experiments, may indeed be influenced by dynamic environmental factors.
The primary contribution of this manuscript is to establish that the FBG-based temperature rise rate is a robust and sensitive precursor to flashover. Determining a baseline threshold under standardized conditions is the critical first step. This baseline serves as the fundamental reference point upon which any future adaptive model must be built. The reviewer's suggestion is directly aligned with our vision for the next phase of this research.
We believe that establishing a physically meaningful precursor (the temperature rise rate) is the most significant breakthrough. Transforming the warning logic from fixed to adaptive is the necessary engineering progression that builds upon this foundation. Our next stage of work is based on the method proposed in this article for online trial operation, establishing corresponding monitoring systems, proposing flashover thresholds under different environments, and analyzing the impact of environmental factors on flashover.
Comment 6: The limited sample size undermines statistical robustness of flashover characteristics across manufacturer variations.
Response 6:
We thank the reviewer for raising this valid point regarding the statistical robustness and generalizability of our findings. We acknowledge that our study, which focused on a specific type of 10kV composite insulator from a single manufacturer, does not provide a broad statistical dataset covering variations across different manufacturers.
Please allow us to clarify the primary objective of our work and how we have addressed this concern in the revised manuscript:
The main goal of this initial study was to establish a fundamental proof of concept. We aimed to demonstrate, for the first time, that the FBG-based temperature rise rate at the interface is a viable and highly sensitive precursor to flashover. This mechanistic investigation required a controlled experimental setup where key variables (like material composition and geometry) were kept constant to isolate and validate the core physical phenomenon. As we added the results of the silicone rubber flat plate test in Section 5.3.1, which provides strong evidence for the underlying principle —the threshold for critical flashover is primarily governed by the properties of the silicone rubber material and the physics of arc heating.
We believe that the validation of the core mechanism is the most critical foundational step. Our work provides the necessary groundwork upon which broader statistical studies, as suggested by the reviewer, can be effectively designed. Thank you for this constructive comment and we will conduct more extensive research in the future to increase the applicability of early warning systems.
Comment 7: The literature review demonstrates adequate engagement with recent advancements in early warning. More publications on application of distributed fibre optic sensors (DOI: 10.1080/10298436.2025.2479645) should be incorporated to strengthen context.
Response 7:
We thank the reviewer for the positive feedback on our literature review and for suggesting the highly relevant reference on the application of distributed fiber optic sensors (DOI: 10.1080/10298436.2025.2479645). We have carefully studied this publication and agree that it provides valuable context for our work. Accordingly, we have incorporated a discussion and citation of this paper in the revised manuscript (e.g., in the Introduction or Literature Review section) to strengthen the background of our study.
Comment 8: How do the authors reconcile discrepancies between laboratory salt-fog conditions and actual field pollution patterns affecting insulator performance?
Response 8:
We thank the reviewer for this critical question, which rightly points the fundamental challenge of simulating complex field conditions in a controlled laboratory environment. We appreciate the opportunity to discuss the rationale behind our choice of the salt-fog test and its relevance to actual field performance.
We fully agree with the reviewer that discrepancies exist. Actual field pollution is a complex mixture of soluble salts (ESDD) and non-soluble materials (NSDD) like dust, ash, or cement, with non-uniform distribution patterns. In contrast, the standard salt-fog test primarily generates a uniform layer of soluble salts, which is a simplification of reality.
Reference [31] in this article mentions that the salt fog test is a well-established and internationally standardized method. Its primary strength lies in its excellent reproducibility, which allows for meaningful and comparable results between different studies. This was essential for our primary goal.
In summary, the coastal salt fog environment under actual working conditions also fluctuates within a certain range, with its biggest characteristics being high humidity and high salt content. This article mainly simulates the working condition in the laboratory to conduct AC pollution flashover tests, providing a certain data basis and theoretical support for future network trial operation. We once again thank the reviewers for their valuable feedback.
Comment 9: How do the authors reconcile discrepancies between laboratory salt-fog conditions and actual field pollution patterns affecting insulator performance?
Response 9:
We thank the reviewer for this insightful comment. We acknowledge that our initial description may not have sufficiently highlighted the adaptive nature of our algorithm. The method proposed in our study is, in fact, an adaptive algorithm. As detailed in the newly added Section 5.3.2, the core of our approach involves a data-driven weighting mechanism that dynamically determines the warning threshold. We have revised the manuscript to more clearly articulate this adaptive calibration process throughout the relevant sections.
Comment 10: Have the authors verified the proposed method’s efficacy across diverse geographical altitudes where atmospheric pressure impacts arc behavior?
Response 10:
We thank the reviewer for raising this important point regarding the impact of geographical altitude and atmospheric pressure on arc behavior. The reviewer is absolutely correct that atmospheric pressure is a key factor influencing arc development and dielectric strength, which could potentially affect the precise warning threshold.
Please allow us to clarify the scope of our current study and our perspective on this issue:
Scope and Focus of the Present Work: The primary objective of this research was to develop and validate the fundamental principle of the early warning method based on FBG and visible images for a specific, high-risk scenario: flashover on composite insulators in coastal regions. These areas are characterized by high salinity and humidity but are typically at low altitudes. Therefore, our experimental design focused on simulating these coastal conditions at standard atmospheric pressure. We will further investigate the correlation between the threshold of critical flashover and pressure in future research.
We thank the reviewer again for this constructive comment, which has helped us better define the current applicability and future potential of our warning system.
Comment 11: The paper fails to address potential signal interference between adjacent FBG sensors in multi-point monitoring configurations.
Response 11:
We thank the reviewer for this excellent technical point regarding the critical issue of cross-talk in multiplexed FBG sensing systems. We appreciate the opportunity to clarify this aspect of our design.
Please allow us to assure the reviewer that potential signal interference was a key consideration in our sensor network design, and it is effectively mitigated through two primary strategies:
1) Our multi-point FBG system is based on Wavelength Division Multiplexing (WDM). Each FBG sensor is fabricated to have a distinct and stable central Bragg wavelength. The demodulator is designed to detect and track the wavelength shift of each sensor within its allocated spectral window. In our setup, the adjacent FBG sensors are spaced with a spectral separation of about 10 nm (as shown in Table. 2 as added), which is significantly larger than the typical wavelength shift (usually less than 1-2 nm) induced by temperature or strain variations in this application. This design fundamentally prevents the spectra of adjacent sensors from overlapping, thereby eliminating the risk of signal interference.
2) The physical placement of the FBG sensors along the insulator was also strategically planned. The distance between adjacent sensor points (45 mm axial separation) is greater than the expected spatial extent of the highly localized thermal anomalies we aim to detect. The close circumferential proximity (18 mm) of the FBG sensors does not lead to mutual interference due to the following reasons: The FBG sensors are pre-embedded at the critical epoxy resin-core rod interface. They are subsequently overlaid with a 3.2-mm-thick layer of RTV silicone rubber. Both the epoxy resin and the RTV coating possess high dielectric strength and act as robust insulating barriers; The core detection mechanism is based on highly localized, transient temperature spikes caused by arc heating. The time scale of this event is short (milliseconds), and the spatial extent of the heat source is very small (millimeters).
We apologize for not explicitly discussing this important design consideration in the original manuscript. In response to the reviewer's comment, we have added a paragraph in Section 5.1. Thank you again for prompting us to include this crucial technical clarification.

Reviewer 3 Report
Comments and Suggestions for Authors
The manuscript “Early Warning of AC Salt Fog Flashover on Composite Insulators Using Fiber Bragg Grating Sensing and Visible Arc Images” proposes a method to monitor the salt fog flashover on composite insulators by combining the signals detected by multiple fiber Bragg gratings (FBGs) for temperature sensing and visible arc image analysis. The method is novel, and the experimental results are convincing. However, a lot of experimental details are missing in the manuscript. I would suggest its publication in Micromachines with the following revisions:
- Both the temperature and the stress will shift the central reflection wavelength of the FBG sensor. In the case of flashover, the FBG’s wavelength shift will sense both the temperature and the stress. In the current manuscript, the stress caused by flashover is ignored. The authors might discuss how will the stress caused by flashover affect the accuracy of temperature sensing? Is the FBG sensor sensing both the temperature and stress altogether?
- Please elaborate the novelties of this paper in Section 1 by comparing with the published research papers.
- Please explain how neff can be calculated in equation (1). Is there a formula for it?
- In Section 3, please indicate what is the optical fiber’s brand and product number, and what is the epoxy brand and product number.
- In line 129, what is the full name of FRP?
- In line 132, what is the brand and product number of the RTV silicone rubber and how is it coated to the FRP rod? In line 138, the silicone rubber is baked for how long at which temperature?
- In Figure 2(a), what is the diameter of the core rod? What is the length of each optical fiber? What is the sensing length of the fiber grating? Are the FBG51, FBG52 and FBG53 gratings long enough to cover the whole perimeter of the core rod? Figure 2(b) should have 3 optical fibers according to Figure 2(a), now there is only one fiber in Figure 2(b).
- Please add a drawing to explain what the parameters are in Table 1. The parameters in Table 1 are not explained in the context, and they are not self-explanatory.
- In Section 4.1, please add the brand and model numbers of the fogging device, infrared thermal imager and camera used in the experiments in Figure 4.
- In line 209, “this table” should be “Table 3”.
- Please add references for equations (5)-(11).
- In line 342, “…graphics card The…” should be “…graphics card. The…”
- In line 364, please give the full name of ESDD.
Author Response
Manuscript ID: 3894034
Title: Early Warning of AC Salt Fog Flashover on Composite Insulators Using Fiber Bragg Grating Sensing and Visible Arc Images
Dear Editor and Reviewers,
Thank you for giving us the opportunity to revise our manuscript. We are very grateful to the editors and reviewers for their insightful comments and suggestions. We have carefully considered all the comments and have revised the manuscript accordingly. Point-by-point responses to the comments are listed below.
- Report 3
Comment 1: Both the temperature and the stress will shift the central reflection wavelength of the FBG sensor. In the case of flashover, the FBG’s wavelength shift will sense both the temperature and the stress. In the current manuscript, the stress caused by flashover is ignored. The authors might discuss how will the stress caused by flashover affect the accuracy of temperature sensing? Is the FBG sensor sensing both the temperature and stress altogether?
Response 1:
We sincerely thank the reviewer for raising this critical and insightful point regarding the cross-sensitivity of FBG sensors to temperature and strain. This is indeed a fundamental consideration in FBG-based sensing, and we appreciate the opportunity to discuss its impact within the context of our flashover early-warning application.
We acknowledge that the wavelength shift during a flashover event is a combined response to both temperature and mechanical stress. In this initial study, our primary focus was on validating the feasibility of using the FBG signal trend as a precursor, with the core indicator being the rate of wavelength change, which is dominantly indicative of the interface temperature rise rate trend.
- Temporal Dominance of Thermal Effect: We posit that during flashover development, the thermal energy deposited by the arc is the dominant, continuous process, whereas any mechanically-induced stress (e.g., from transient shockwaves) is likely to be brief and impulsive. The sustained and rapidly increasing wavelength drift trend captured by the FBG is therefore primarily attributable to thermal accumulation. A transient stress spike may cause a momentary artifact in the signal but would not generate the consistent upward trend in the wavelength shift rate that is characteristic of impending flashover.
- Physical Buffering: Furthermore, the FBG sensors are pre-embedded within the epoxy resin-RTV silicone rubber interface. These encapsulating materials provide a certain degree of mechanical buffering, which can help attenuate high-frequency stress pulses.
- Robustness of the Warning Logic: Our proposed warning strategy is based on identifying a sustained exceeding of a threshold rate of change. This methodology is inherently robust against sporadic, non-thermal interference. As long as the stress contribution does not systematically mimic the signature of a continuous thermal ramp-up, it will not compromise the reliability of the warning.
Manuscript Revision: The reviewer's suggestion is highly valuable for advancing this technology. We have revised the manuscript to include a discussion on these issues in Section 2. We explicitly state that focusing on the temperature parameter to establish the warning model in this research is reasonable and reliable.
We thank the reviewer again for this constructive comment, which has significantly strengthened the technical depth of our discussion.
Comment 2: Please elaborate the novelties of this paper in Section 1 by comparing with the published research papers.
Response 2:
We thank the reviewer for this valuable suggestion. We have revised Section 1 to elaborate on the novelties of this paper through explicit comparisons with published research, as follows:
- Novelty in Warning Principle:
Existing early-warning methods predominantly rely on a single physical parameter (e.g., leakage current, partial arc, or a single image feature), which are susceptible to interference and offer limited reliability. The core innovation of this paper is the first proposal of a dual-parameter cooperative warning mechanism based on the FBG interfacial temperature rise rate and visible-light image characteristics. By fusing the FBG signal, which reflects the internal thermal dynamics, with image data that captures the external arc's optical characteristics, this mechanism achieves cross-verification and collaborative perception of the flashover process from both internal and external dimensions, fundamentally enhancing the accuracy and reliability of the warning.
- Novelty in Technical Approach:
Regarding the application of FBG, previous works have mostly utilized it for structural strain monitoring or as externally attached temperature sensors. This paper innovatively embeds the FBG sensors at the critical interface of composite insulators (the epoxy resin core rod-silicone rubber interface), transforming them into in-situ, integrated interface thermal state probes. This allows us, for the first time, to directly capture the microscopic transient temperature rise at the critical interface induced by local arcs prior to flashover—a precursor that is elusive to traditional methods, thereby providing a more advanced and fundamental physical criterion for warning.
- Novelty in Warning Model:
Unlike traditional warning models employing fixed thresholds, this paper introduces a data-driven, adaptive threshold calculation framework based on the entropy weight method. This model dynamically assigns weights and calculates thresholds based on the objective characteristics of real-time monitoring data, significantly enhancing the adaptability and robustness of the early-warning system under complex and variable operating conditions.
Direct Comparison with Relevant Studies: Specifically, compared to published research on optical fiber composite insulators or visible-light image analysis, the novelty of this work lies in the deep integration and complementary use of the two technologies, rather than their simple parallel application. Existing literature tends to focus either on optical fiber sensing or image analysis, with data remaining isolated. This paper not only employs both techniques but, more importantly, establishes a cooperative decision-making logic where "FBG interfacial temperature rise triggers the alert, and visible light arc characteristics provide verification". This directly addresses the inherent limitations of single-method monitoring—such as visual blind spots in imaging and potential non-thermal interference in FBG signals—thus constituting a substantial methodological innovation.
We sincerely thank the Reviewer for this valuable suggestion. The comment has helped us to more clearly articulate the core methodological innovation of this work—the deep integration of FBG interfacial thermal sensing and visible-light imaging within a cooperative warning framework. The revised introduction now includes a direct comparison with prior single-parameter approaches, explicitly highlighting this significant advancement.
Comment 3: Please explain how neff can be calculated in equation (1). Is there a formula for it?
Response 3:
We thank the reviewer for this question regarding the detail of the equation.
The effective refractive index neff of a FBG is an intrinsic parameter of the optical fiber waveguide itself, describing the equivalent refractive index for light propagating in the fiber core. This parameter is primarily determined by the fiber material (e.g., germanium-doped silica) and waveguide structure (e.g., core/cladding geometry and refractive index profile). Its value is typically provided as a nominal specification by the fiber manufacturer or can be derived from mode-solving theory. In our study, the effective refractive index neff for the employed FBG sensors is a fixed constant, and its value was taken directly from the product datasheet. Therefore, in Equation (1), we treated it as a known parameter without elaborating on its detailed calculation in the manuscript.
Following the reviewer's suggestion, we added a clarification in Section 2 regarding this and cite the source of its specific value.
Comment 4: In Section 3, please indicate what is the optical fiber’s brand and product number, and what is the epoxy brand and product number.
Response 4:
We thank the reviewer for this comment, which helps to improve the clarity and reproducibility of our work. The requested details regarding the materials used in our experiments are provided below:
- Optical Fiber: The FBG sensors were customized by Shenzhen Zhongke Sensing Technology Co., Ltd. These fibers meet all the necessary specifications for our application, with standard parameters including a core diameter of 9 µm and a cladding diameter of 125 µm.
- Epoxy Resin: The epoxy used for embedding the FBG sensors at the core rod interface is the Kafuter K-703 two-part, high-performance epoxy adhesive. This product was selected for its excellent electrical insulation properties, with a dielectric strength of ≥15 kV/mm, and a service temperature range of -40°C to 200°C, making it well-suited for the intended operational environment.
We have revised Section 3 of the manuscript to include these specifications to ensure full transparency.
Comment 5: In line 129, what is the full name of FRP?
Response 5:
We sincerely thank the reviewer for pointing out our oversight in not spelling out the FRP acronym at its first occurrence.
FRP stands for "Fiber-Reinforced Polymer" (also commonly referred to as Fiber-Reinforced Plastic). In the specific context of composite insulators for high-voltage applications, it denotes the fiberglass-reinforced epoxy resin that constitutes the core rod, which is the primary load-bearing component.
We have revised in the manuscript to include this full name and clarification upon its first occurrence. This suggestion has been very helpful in improving the clarity of our manuscript.
Comment 6: In line 132, what is the brand and product number of the RTV silicone rubber and how is it coated to the FRP rod? In line 138, the silicone rubber is baked for how long at which temperature?
Response 6:
We thank the reviewer for these specific questions regarding the materials and processes in our experimental setup. Please find our detailed responses below, which have been incorporated into the revised Section 3 of the manuscript.
Regarding Line 132 (RTV Coating): The RTV silicone rubber used is Kafuter K-704. It was applied to the surface of the FRP rod using a manual dispensing gun to ensure a thin and uniform coating that fully covered and encapsulated each embedded optical fiber. This initial step ensures the integrity of the fiber-optic sensors before proceeding with the standard manufacturing process for composite insulators.
Regarding Line 138 (Baking Process): Following the RTV application, and consistent with the standard manufacturing process for conventional composite insulators using High-Temperature Vulcanizing (HTV) silicone rubber encapsulation, the core rod is baked at a temperature of 140 °C for a duration of over 30 minutes. This process was designed and verified to ensure that the core rod itself reached a temperature of 100 °C. After the subsequent manufacturing is completed, ensure that the optical fiber is not damaged.
We appreciate the reviewer's attention to detail, which has helped us improve the technical clarity and reproducibility of our methodology section.
Comment 7: In Figure 2(a), what is the diameter of the core rod? What is the length of each optical fiber? What is the sensing length of the fiber grating? Are the FBG51, FBG52 and FBG53 gratings long enough to cover the whole perimeter of the core rod? Figure 2(b) should have 3 optical fibers according to Figure 2(a), now there is only one fiber in Figure 2(b).
Response 7:
We sincerely thank the reviewer for their meticulous examination of our figures and for these insightful questions, which have helped us improve the clarity and accuracy of our manuscript. Please find our point-by-point responses below, and note that we have revised Figures 2(a) and Figures 2(b) accordingly.
- Diameter of the core rod: The diameter of the Fiber-Reinforced Polymer (FRP) core rod is 18 mm, as now explicitly stated in Table 1.
- Length of each optical fiber: The total length of each optical fiber used is 3.5 meters. This includes a 3-meter-long pigtail reserved for connection to the optical sensing interrogator during experiments.
- Sensing length of the fiber grating: The sensing length of each Fiber Bragg Grating (FBG) is 10 mm. This parameter has been added to the revised Section 2.1 (Sensor System Design).
- Coverage of the core rod's perimeter by FBGs: The reviewer raises an excellent point regarding circumferential coverage. With a core rod diameter of 18 mm (circumference ≈ 56.5 mm) and an FBG sensing length of 10 mm, a single FBG cannot cover the entire perimeter. The FBG sensors (FBG51, FBG52, FBG53) are strategically placed at specific, critical radial positions (e.g., at 0°, 120°, and 240° around the circumference) to monitor localized thermal activity at these points. They are not intended to form a continuous sensing ring. We have clarified this intent in the revised figure caption and text. Furthermore, we have re-drawn Figure 2(a) to avoid the previous misinterpretation that FBG51-53 were concentrically located; the new figure clearly shows their distinct radial positions, with red annular areas indicating the shed locations.
- Number of fibers in Figure 2(b): We thank the reviewer for identifying this inconsistency. Figure 2(b) has been re-drawn to accurately depict the three separate optical fibers corresponding to the sensor layout in Figure 2(a), providing a clearer representation of the spatial distribution of the FBGs.
We believe the revised figures and text now more accurately and clearly represent our experimental setup. We are grateful for the reviewer's valuable feedback, which has significantly strengthened our paper.
Comment 8: Please add a drawing to explain what the parameters are in Table 1. The parameters in Table 1 are not explained in the context, and they are not self-explanatory.
Response 8:
We thank the reviewer for this valuable suggestion. We agree that a visual aid will significantly improve the clarity of the parameters presented in Table 1.
In response, we have added a new schematic diagram as Figure 3 (b) to the manuscript. This new figure explicitly illustrates the structural geometry of the fiber-optic composite insulator and provides clear definitions for all parameters listed in Table 1.
We believe this addition effectively resolves the ambiguity and greatly enhances the reader's understanding of our insulator design.
Comment 9: In Section 4.1, please add the brand and model numbers of the fogging device, infrared thermal imager and camera used in the experiments in Figure 4.
Response 9:
We thank the reviewer for this suggestion to improve the reproducibility of our experimental setup. We have revised Section 4.1 to include the brand and model numbers of the key equipment used, as follows:
Fogging Device: The ultrasonic atomization device model is JY-9.0F.
Infrared Thermal Imager: The thermal data for comparative analysis was captured using a Hikvision TP-96 infrared camera.
Camera (Visible light): The visible-light videos of the arc were recorded with a Lenovo X5Q.
This information has been added to the revised manuscript in Section 4.1. We believe this clarification enhances the technical detail and allows for better comparison with future studies.
Comment 10: In line 209, “this table” should be “Table 3”.
Response 10:
We thank the reviewer for pointing out this oversight. The phrase "this table" has been corrected to "Table 3" in the revised manuscript to ensure precise and unambiguous referencing.
Comment 11: Please add references for equations (5)-(11).
Response 11:
We appreciate the reviewer's question regarding references for Equations (5)-(11). These equations constitute the original calculation process of the data-driven adaptive threshold method proposed in this paper. They detail the step-by-step procedure, from data normalization to the final weight determination using the entropy weight method, which was specifically applied to our unique set of multi-parameter features (three characteristics of arc visible images). Therefore, they are presented here for the first time as a novel component of our methodology. We have clarified this in the revised manuscript to avoid any potential misunderstanding.
Comment 12: In line 342, “…graphics card The…” should be “…graphics card. The…”
Response 12:
We sincerely thank the reviewer for their meticulous review and for identifying this typographical error. The missing period has been corrected in the revised manuscript. The sentence now correctly reads: "...graphics card. The..."
Comment 13: In line 364, please give the full name of ESDD.
Response 13:
We thank the reviewer for this helpful remark. The full name of "ESDD" is Equivalent Salt Deposit Density. We have revised in the manuscript to include the full term upon its first appearance, which is a standard and crucial parameter for quantifying pollution severity on insulators.
We sincerely thank the reviewers for their thoughtful and constructive comments. These suggestions have been invaluable in helping us to improve the clarity, rigor, and overall quality of our manuscript.
Sincerely,
Xiaoxiang Wu
School of Electric Power Engineering, South China University of Technology
Oct 13, 2025
Round 2
Reviewer 1 Report
Comments and Suggestions for Authors
The authors have addressed my concerns.
Author Response
Thank you!
Reviewer 2 Report
Comments and Suggestions for Authors
The manuscript has been revised in accordance with the reviewer's suggestions. I recommend its acceptance for publication.
Author Response
Thank you!
Reviewer 3 Report
Comments and Suggestions for Authors
I appreciate the authors’ detailed explanation and improvement on the revised manuscript. All of my comments were addressed. I would suggest the following minor revisions before the publication of this paper on Micromachines:
- Please add the length of the optical fiber in the manuscript. In the reply to my review comments, the authors mentioned they are 3.5-meter long including “a 3-meter pigtail reserved for connection to the optical sensing interrogator during experiments”. On the core rod, the fiber is already longer than two times of the shade spacing which in total is 0.9-meter long. With a 3-meter pigtail, the total length will be more than 3.5 meters. Please clarify this in the manuscript.
- Please remove the words “, the core rod is baked” in line 162.
- The authors introduced the infrared thermal imager. In the reply to my review comments, the authors replied, “Infrared Thermal Imager: The thermal data for comparative analysis was captured using a Hikvision TP-96 infrared camera”. However, in the manuscript, I did not find any data obtained from the infrared camera being used for discussion. Is this thermal camera important for this research?
Author Response
Thank you for giving us the opportunity to revise our manuscript. We are very grateful to the editors and reviewers for their insightful comments and suggestions. We have carefully considered all the comments and have revised the manuscript accordingly. Point-by-point responses to the comments are listed below.
Comment 1: 1.Please add the length of the optical fiber in the manuscript. In the reply to my review comments, the authors mentioned they are 3.5-meter long including “a 3-meter pigtail reserved for connection to the optical sensing interrogator during experiments”. On the core rod, the fiber is already longer than two times of the shade spacing which in total is 0.9-meter long. With a 3-meter pigtail, the total length will be more than 3.5 meters. Please clarify this in the manuscript.
Response 1:
We sincerely thank the reviewer for the thorough review and valuable comments. The issue regarding the inconsistent description of the optical fiber length is very pertinent, and we apologize for any confusion caused by the incomplete information in our previous response.
The following clarifications are provided based on the actual experimental setup:
Exact Length: The total length of the customized optical fiber we used is approximately 3.5 meters. This includes approximately 0.29 meters of sensing fiber embedded at the core rod-silicone rubber sheath interface, and over 3.0 meters of pigtail used for connection.
Explanation of Length Discrepancy: The length of the optical fiber within the interface is greater than 90 millimeters (0.09 meters), approximately equal to the total structural height of the composite insulator, which is about 290 millimeters (0.29 meters). Our previous description lacked precision, leading to this misunderstanding.
Rationale for Length and Safety Consideration: The provision of a pigtail exceeding 3 meters is primarily to ensure operational safety by allowing the optical sensing interrogator to be placed at a safe distance during high-voltage experiments.
Following the reviewer's suggestion, we have now added a precise description and clarification of the optical fiber length in the revised manuscript (Line 149, Section 3). The modified text reads:
"The customized optical fiber with a total length of approximately 3.5 meters was used. Among this, about 0.29 meters was precisely positioned within the core rod-silicone rubber sheath interface of the composite insulator, serving as the strain sensing region, while the remaining approximately 3.21 meters served as the pigtail to ensure the measurement equipment could be located in a safe zone during high-voltage experiments."
Comment 2: 2.Please remove the words “, the core rod is baked” in line 162..
Response 2:
We thank the reviewer for pointing out this oversight. The words ", the core rod is baked" have been removed as suggested.
Comment 3: 3.The authors introduced the infrared thermal imager. In the reply to my review comments, the authors replied, “Infrared Thermal Imager: The thermal data for comparative analysis was captured using a Hikvision TP-96 infrared camera”. However, in the manuscript, I did not find any data obtained from the infrared camera being used for discussion. Is this thermal camera important for this research?
Response 3:
We thank the reviewer for their thorough review and valuable comments. The issue raised is indeed critical, and we acknowledge this oversight. The reviewer is absolutely correct. The infrared thermal imager was not utilized in this study.
In our previous research, we discovered that infrared thermal imaging technology is more suitable for in-depth analysis of thermal effects and mechanisms during discharge, and it exhibits certain latency in presenting surface temperature. However, the core objective of this research is to develop an early flashover warning method based on optical fiber sensing, rather than investigating the thermal mechanisms of discharge. In our comparative studies, we found that for the specific goal of flashover warning, the response of the fiber sensing method is more direct and pronounced than that of the thermal imager.
Therefore, to ensure the focus and rigor of the manuscript's content and core argument, we have removed“. The model of infrared thermal imager is Hikvision TP-96” in the revised version. Accordingly, the description of the experimental setup and methodology has been refined to focus solely on the fiber sensing system actually used in this study.
We thank the reviewer for their keen insight, which has been greatly helpful in improving the quality of this paper.
We sincerely thank the reviewers for their thoughtful and constructive comments again. These suggestions have been invaluable in helping us to improve the clarity, rigor, and overall quality of our manuscript.